# How childhood psychological abuse affects adolescent cyberbullying: The chain mediating role of self-efficacy and psychological resilience

**Haihua Ying** [1]*, **Yang Han** [2]

**1** School of Education Science, Nanjing Normal University, Jiangsu, China, **2** School of Computing, Nanjing University of Information Science & Technology, Jiangsu, China

* 20120620@hhu.edu.cn

## Abstract

### Background

Despite the recognition of the impact of childhood psychological abuse, self-efficacy, and psychological resilience on cyberbullying, there is still a gap in understanding the specific mechanisms through which childhood psychological abuse impacts cyberbullying via self-efficacy and psychological resilience.

### Methods

Based on the Social Cognitive Theory, this study aims to investigate the link between childhood psychological abuse and cyberbullying in adolescents, mediated by the sequential roles of self-efficacy and psychological resilience. The sample consisted of 891 students ($M = 15.40$, $SD = 1.698$) selected from four public secondary schools in Jiangsu Province, Eastern China. All the participants filled in the structured self-report questionnaires on childhood psychological abuse, self-efficacy, psychological resilience, and cyberbullying. The data were analyzed using SPSS 24.0 and structural equation modeling (SEM) in AMOS 24.0.

### Results

The findings of this study are as follows: (1) Childhood psychological abuse is positively associated with adolescent cyberbullying; (2) Self-efficacy plays a mediating role between childhood psychological abuse and adolescent cyberbullying; (3) Psychological resilience plays a mediating role between childhood psychological abuse and adolescent cyberbullying; (4) Self-efficacy and psychological resilience play a chain mediation role between childhood psychological abuse and adolescent cyberbullying.

### Conclusion

This study contributes to a deeper understanding of the underlying mechanisms linking childhood psychological abuse to adolescent cyberbullying, shedding light on potential

**Data Availability Statement:** All relevant data are within the manuscript and its Supporting information files.

**Funding:** The author(s) received no specific funding for this work.

**Competing interests:** The authors have declared that no competing interests exist.

pathways for targeted interventions and support programs to promote the well-being of adolescents in the face of early adversity.

## 1. Introduction

The rapid development of the internet has brought many conveniences to our lives, but it has also brought numerous negative impacts, such as internet addiction [1], online fraud [2], and cyberbullying [3]. Among these, cyberbullying has been referred to as an "invisible fist", with its harm being greater than traditional bullying and having a wider impact [4]. Cyberbullying is characterized by deliberate, repetitive, and malicious acts which are carried out using modern communication technologies, aimed at causing harm to others [5, 6]. It comprises two dimensions: cyberbullying victimization and cyberbullying perpetration [7]. This pervasive issue is recognized globally [8], as evidenced by data from 2019, which revealed that one-third of young people from 30 countries consistently reported being victims of cyberbullying [9]. In China, the number of underage internet users reached 183 million in 2020, with 24.3% of minors reporting experiencing cyber violence, according to the "Research Report on Internet Usage among Minors in China in 2020" [10]. Adolescents are particularly vulnerable to cyberbullying [11]. The survey results indicate that approximately 52.2% of adolescents in China have experienced at least one incident of cyberbullying in the past year [12]. Cyberbullying not only impacts the psychological well-being of adolescents, but also lead to their difficulties in social adaptation and potentially tragic outcomes [13]. Therefore, it is of great significance to explore the factors influencing adolescent cyberbullying for prevention and intervention.

Cyberbullying is influenced by both environmental factors and individual factors [14]. Childhood psychological abuse is an important environmental factor influencing cyberbullying [15]. Child psychological abuse refers to the series of inappropriate fostering methods that are repeatedly and continuously adopted by the fosterer during the process of children's growth, including intimidation, neglect, disparagement, interference, and indulgence [16]. Previous research has established a positive correlation between childhood psychological abuse and adolescent cyberbullying [17, 18]. High levels of childhood psychological abuse have been associated with higher levels of cyberbullying, while low levels of childhood psychological abuse can hinder adolescent cyberbullying [19]. Self-efficacy and psychological resilience are two individual factors that have been extensively explored in relation to cyberbullying [20]. Self-efficacy refers to an individual's confidence and expectation in their ability to take effective action and accomplish tasks in specific situations [21]. Psychological resilience is defined as the adaptive ability to maintain an active life despite adversity and stressful events [22]. They have been found to exhibit a negative correlation with adolescent cyberbullying. For example, Özdemir and Bektaş suggested that self-efficacy plays a negative role in cyberbullying [23]. Similarly, Clark and Bussey observed a noteworthy negative association between self-efficacy and cyberbullying among adolescents [24]. Güçlü-Aydogan et al. posited that psychological resilience has a negative impact on cyberbullying [20]. The findings highlight the importance of considering both self-efficacy and psychological resilience in understanding adolescent cyberbullying.

Despite scholars proposing the influence of these factors on adolescent cyberbullying, the specific mechanisms through which childhood psychological abuse affects adolescent cyberbullying via self-efficacy and psychological resilience remain understudied. To address this research gap, this study aims to investigate the interactive effects of childhood psychological

abuse, self-efficacy, psychological resilience on adolescent cyberbullying, thereby providing a holistic understanding of the relationship between these factors. Furthermore, the study endeavors to investigate the impact of childhood psychological abuse on adolescent cyberbullying, with a specific focus on the mediating roles of self-efficacy and psychological resilience. This study seeks to address the following questions: First, what is the relationship between childhood psychological abuse and adolescent cyberbullying? Second, does self-efficacy mediate the relationship between childhood psychological abuse and adolescent cyberbullying? Third, does psychological resilience mediate the relationship between childhood psychological abuse and adolescent cyberbullying? Fourth, is there a serial mediation effect of self-efficacy and psychological resilience between childhood psychological abuse and adolescent cyberbullying? This study is significant as it addresses a gap in the existing literature and provides insights into the determinants of adolescent cyberbullying. Moreover, by exploring the mediating mechanisms through which childhood psychological abuse impacts adolescent cyberbullying, this study provides valuable guidance for educators and parents seeking to reduce adolescent cyberbullying.

The structure of the remaining sections of this article is as follows. Section 2 provides an overview of the theoretical background and hypothesis development. Section 3 details the materials and methods, encompassing participants, the research process, research instruments, and statistical analysis. Section 4 covers common method variance, descriptive statistics, correlation analysis, examination of the model, and testing for mediation effects. Section 6 presents the findings, limitations, and implications.

## 2. Theoretical background and hypothesis development

### 2.1 Theoretical background

Social Cognitive Theory (SCT), originally proposed by Bandura [21], provides a robust theoretical framework for this study. The theory includes three elements: environment, personal factors, and behavior [25]. Environment is defined as the external influences that affect an individual's behavior, such as social norms, cultural values, and physical surroundings, while personal factors refer to an individual's cognitive, affective, and biological characteristics, including beliefs, emotions, and genetic predispositions [26]. Behavior encompasses the actions and responses exhibited by an individual in various situations [21]. Unlike some other theories that focus solely on either environmental or personal determinants of behavior, SCT emphasizes the dynamic interaction between environment, personal factors, and behavior. It posits that individuals are not simply passive recipients of environmental influences, but rather they actively engage with and interpret their surroundings. Personal factors, such as cognitive processes and emotional states, play a crucial role in mediating the impact of the environment on behavior. Similarly, an individual's behavior can also influence and modify their environment and personal factors. In this study, childhood psychological abuse is considered an environmental factor, while self-efficacy and psychological resilience as two personal factors. Cyberbullying, heralded as individuals' social behavior, can also be explained by environmental and personal factors [27]. Childhood psychological abuse has a significant impact on the development of individuals' self-efficacy. An enhanced sense of self-efficacy enables individuals to effectively cope with academic and social challenges, engage actively in demanding learning tasks, and develop psychological resilience [28]. Moreover, self-efficacy significantly reduces the occurrence of cyberbullying by bolstering individuals' confidence and coping abilities, while psychological resilience lowers the risk of becoming a victim of cyberbullying by improving individuals' adaptability to adversity [20]. By employing this theoretical framework, we can gain a comprehensive understanding of the association between childhood

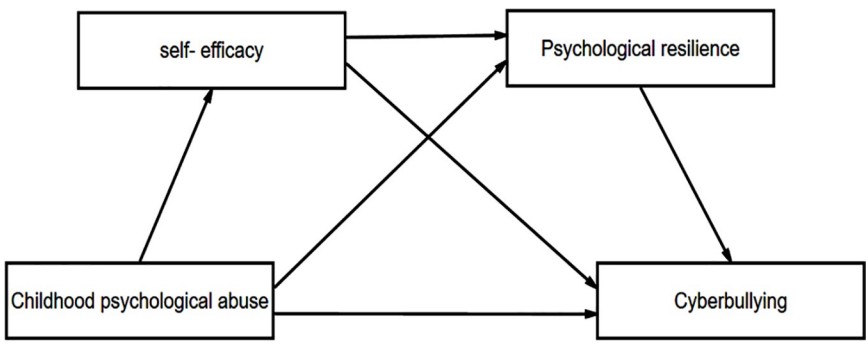

**Fig 1. The theoretical model.**

psychological abuse and cyberbullying, elucidating the mediating roles of self-efficacy and psychological resilience. This theoretical model in the study is visually represented in Fig 1.

## 2.2 Hypothesis development

**2.2.1 Childhood psychological abuse and cyberbullying.** Numerous studies have provided compelling evidence of the link between childhood psychological abuse and subsequent engagement in cyberbullying behaviors [15, 29, 30]. Research has proposed that adverse experiences of psychological abuse in childhood can impact brain function states, such as persistent stress and heightened neurotic anxiety, prompting individuals to suppress and bury these feelings in their subconscious, ultimately leading to engaging in cyberbullying behavior [31]. Research has also proposed that childhood psychological abuse can have an impact on psychological development, thus leading to cyberbullying [19, 32]. For instance, Xu and Zheng demonstrated that childhood emotional abuse can damage an individual's self-esteem and self-confidence, making them seek to control and gain a sense of power through cyberbullying [33]. Moreover, Li et al. identified that childhood psychological abuse may lead to inner feelings of anger in individuals, causing them to seek comfort and escape from reality in online environments, ultimately leading them to release these negative emotions by bullying others online [34]. Based on the evidence presented in the literature, it is hypothesized:

H1: Childhood psychological abuse is positively associated with adolescent cyberbullying.

**2.2.2 Self-efficacy as a mediator.** There is a well-established negative relationship between childhood psychological abuse and self-efficacy [35]. For example, Soffer et al. conducted a study that revealed individuals who experienced childhood psychological abuse reported lower levels of self-efficacy in various domains, such as academic, social, and personal domains [36]. This suggests that the negative experiences associated with abuse can undermine an individual's belief in their capabilities. Supporting this notion, Hosey emphasized the detrimental effects of childhood psychological abuse on an individual's self-efficacy beliefs [37]. Their research highlighted the long-lasting impact of abuse on self-efficacy. Furthermore, Bentley and Zamir conducted a longitudinal study that found the negative relationship between childhood psychological abuse and self-efficacy persisted over time [38]. This suggests that the effects of abuse on self-efficacy may endure throughout adolescence and beyond. Taken together, these studies provide compelling evidence that childhood psychological abuse can significantly impact an individual's self-efficacy.

Studies have explored the relationship between self-efficacy and cyberbullying [23, 39]. Clark and Bussey conducted a study examining the relationship between self-efficacy and cyberbullying victimization and revealed that higher levels of self-efficacy were associated with higher rates of defending behavior during cyberbullying episodes [24]. Similarly, Bussey et al. investigated the relationship between self-efficacy and cyberbullying defending and indicated that individuals with a high level of self-efficacy were more likely to defend cyberbullying [40]. Ferreira et al. surveyed 676 students from the fifth to twelfth grade and found that self-efficacy significantly impacted cyberbullying behavior, with students exhibiting higher self-efficacy demonstrating more proactive problem-solving behavior, thereby reducing instances of cyber-bullying [41]. Additionally, Ybarra and Mitchell found that self-efficacy plays a crucial role in moderating the negative effects of cyberbullying [42]. Their studies revealed that individuals with higher self-efficacy were better able to cope with and overcome the negative consequences of cyberbullying.

The above views indicate that childhood psychological abuse may negatively affect individuals' self-efficacy, which in turn, may contribute to an increased likelihood of engaging in cyberbullying behavior. Based on these, the following assumption is proposed:

H2: Self-efficacy may play a mediating role in the association between childhood psychological abuse and adolescent cyberbullying.

**2.2.3 Psychological resilience as a mediator.** It has been found that psychological resilience can be influenced by childhood psychological abuse [43]. Yang et al. carried out a cross-sectional survey among 1607 adolescents and proposed that childhood psychological abuse may contribute to the development of psychological resilience during the learning process [44]. Additionally, Arslan conducted a survey involving 937 adolescents from various high schools and emphasized that childhood psychological abuse was a consistent predictor of psychological resilience [45]. These findings collectively support the notion that childhood psychological abuse may have a positive impact on the psychological resilience of adolescents.

Studies have shown that psychological resilience can influence cyberbullying [46, 47]. Students with higher levels of resilience were less likely to engage in cyberbullying behaviors [48]. Hinduja and Patchin have argued that students with more psychological resilience were less likely to report being online victims, and among those who did report being victims, their psychological resilience worked as a "buffer," preventing negative effects at school [49]. Similarly, Güçlü-Aydogan et al. investigated the role of psychological resilience in mitigating the impact of cyberbullying and found adolescents who exhibit higher levels of psychological resilience are capable of surviving adversity and uncertainty through the use of healthy, effective, and adaptable coping mechanisms, which may result in reduced cyber victimization [20]. Zhang et al. have demonstrated that students who experienced more childhood psychological abuse have lower psychological resilience, which plays a crucial role in bullying victimization [50]. Therefore, this study speculates that there is a positive relationship between adolescents' psychological resilience and their cyberbullying, and psychological resilience may play an intermediary role between childhood psychological abuse and cyberbullying.

Psychological resilience is believed to be influenced by self-efficacy [51]. Bandura proposed a comprehensive framework for understanding the role of self-efficacy in promoting psychological resilience [21]. Individuals with higher levels of self-efficacy are better equipped to navigate and overcome challenges, leading to greater psychological resilience [52]. Sabouripour et al. [28] revealed that individuals with higher levels of self-efficacy demonstrated greater psychological resilience when facing health challenges. Therefore, it is believed that childhood

psychological abuse may influence cyberbullying via the serial variables of self-efficacy and psychological resilience. Given this, the following hypotheses are proposed:

H3: Psychological resilience plays a mediating role in the association between childhood psychological abuse and adolescents' cyberbullying.

H4: Self-efficacy and psychological resilience play a chain mediating role in the association between childhood psychological abuse and adolescent cyberbullying.

Based on Social Cognitive Theory and the above hypotheses, this study aims to apply SCT to explore the relationship between childhood psychological abuse and adolescents' cyberbullying. Specifically, we will examine the mediating roles of self-efficacy and psychological resilience. A theoretical model (Fig 1) will be constructed to investigate these relationships.

## 3. Materials and methods

### 3.1 Participants

This study utilized G*power 3.1 software [53] to calculate the required sample size, with an effect size set at 0.3 and α set at 0.05. The results indicated that in order to achieve a statistical power of 0.95, a total of 145 participants were needed. Furthermore, based on the requirement of Structural Equation Modeling (SEM) [54] that the appropriate sample size should be at least ten times the total observed variables, it was determined that a minimum of 800 participants would be necessary. The survey initially identified schools for sample collection based on convenience sampling principles. However, to ensure representativeness, cluster sampling was subsequently employed at the class level to select the 1,000 samples from 4 secondary schools (2 public junior high schools and 2 public senior high schools) in Jiangsu province, China. The selected public schools for this study exhibit diversity in terms of student backgrounds, academic achievements, and socio-economic statuses, thereby approximating the overall student population in the region. A total of the 1000 questionnaires were distributed, and after excluding the invalid questionnaires with missing answers or consistent responses, 891 valid questionnaires were collected, resulting in an effective response rate of 89.1%. Participants were aged 13 to 18 years old (M = 15.40, SD = 1.698), with 408 (45.8%) being boys, and 483 (54.2%) being girls. In terms of grade, the participants included 152 (17.1%) in the 7th grade, 167 (18.7%) in the 8th grade, 148 (16.6%) in the 9th grade, and 164 (18.4%) in the 10th grade, 113 (12.7%) in the 11th grade, 147 (16.5%) in the 12th grade.

### 3.2 Procedure

The study was conducted in accordance with the approved guidelines from the Ethical Review Committee of Hohai University (Protocol Number: Hhu10294-240125). Additionally, consent was obtained from the principals, students, and their parents in the participating schools. Before the survey, students were informed about the confidentiality of the survey results and their intended use solely for research purposes in class. They were also assured that measures had been implemented to safeguard their privacy. The questionnaires were then distributed and thoroughly explained to the participants. After 15 minutes, the trained research assistants collected the questionnaires on the spot, and subsequently, the data from the questionnaires were meticulously sorted and analyzed to derive meaningful conclusions.

### 3.3 Research instrument

**3.3.1 Childhood psychological abuse scale.** The measurement of childhood psychological abuse was conducted using Pan et al.'s scale [16], which comprises 23 items capturing five

dimensions: intimidation, neglect, disparagement, interference, and indulgence. For example, one item on the scale is "My parents interrogate me about the details of my interactions with friends." A 5-point Likert scale was employed, with scores ranging from 0 to 4, indicating "none" to "always", and higher scores reflecting higher childhood psychological abuse. The scale has been demonstrated to possess good reliability and validity [55].

**3.3.2 Self-efficacy scale.** Self-efficacy was measured using the scale developed by Wang et al. [56], which is based on Schwarzer and Jerusalem's General Self-Efficacy Scale [57]. This scale consists of 10 items, presented in a single structure, with statements such as "I can calmly face challenges because I trust my ability to handle problems." A 4-point Likert scale was utilized, with scores ranging from 1–4, representing "strongly disagree" to "strongly agree" respectively. Higher scores indicate higher levels of self-efficacy. The scale has good reliability and validity in previous study [58].

**3.3.3 Psychological resilience scale.** The psychological resilience scale, developed by Hu and Gan [59], was utilized to evaluate the psychological resilience levels of adolescents. This scale comprises 27 items, encompassing five dimensions: goal focus, emotional control, positive cognition, interpersonal assistance, and family support. For example, one item states, "I believe that everything has its positive aspects". The scale is rated on a 5-point Likert scale, with scores ranging from 1(strongly disagree) to 5(strongly agree), and higher scores indicating a stronger sense of psychological resilience. The scale demonstrates good reliability and validity, which has been validated by Xiao et al. [60].

**3.3.4 Cyberbullying scale.** The measurement of adolescents' cyberbullying was carried out using the revised Chinese version of the Cyberbullying Scale by You [7]. This scale comprises two subscales: the cyberbullying victimization scale (12 items, such as "Someone has shared or used my photos or videos online without my consent") and the cyberbullying perpetration scale (8 items, such as "When conversing with someone online and things don't go my way, I may resort to using offensive language to insult them"). The scale utilizes a 4-point rating, ranging from 1 (Never happened) to 4 (Frequently happened), with higher scores indicating a higher frequency of cyberbullying. Studies have demonstrated good reliability and validity among Chinese adolescents [61, 62].

## 3.4 Statistical analysis

The collected data were analyzed using SPSS 24.0 and AMOS 24.0. Initially, the Harman single-factor test was conducted in SPSS 24.0 to assess common method variance. Subsequently, correlation analysis was performed on the variables of childhood psychological abuse, self-efficacy, psychological resilience, and cyberbullying in SPSS 24.0. Then, the measurement model and structural model were assessed using factor loadings, Cronbach's α, CR, AVE, and goodness-of-fit. Finally, the mediation test was conducted utilizing AMOS 24.0. To ascertain the statistical significance of the mediating effects posited by the hypotheses, a bootstrapping method was employed, with the generation of 95% confidence intervals to provide a robust evaluation of these effects.

## 4. Results

### 4.1 Common method bias analysis

To mitigate the influence of common method bias, in addition to ensuring anonymous responses during the survey, Harman's single-factor test was conducted [63]. Exploratory factor analysis was performed on the 80 items of the questionnaire, and an unrotated principal component analysis revealed the presence of 11 factors with eigenvalues greater than 1.

**Table 1. Descriptive statistics and correlation analysis (N = 891).**

| Varibles | M | SD | 1 | 2 | 3 | 4 |
|---|---|---|---|---|---|---|
| 1 Childhood psychological abuse | 0.505 | 0.634 | 1 | | | |
| 2 Self-efficacy | 2.734 | 0.584 | -0.162** | 1 | | |
| 3 Psychological resilience | 3.514 | 0.949 | -0.445** | 0.459** | 1 | |
| 4 Cyberbullying | 1.514 | 0.813 | 0.398** | -0.309** | -0.490** | 1 |

Note:

*$p < 0.05$,

**$p < 0.01$,

***$p < 0.001$, same below.

However, the first factor accounted for only 32.534% of the variance, which is below the critical threshold of 40% [64], indicating that there is no significant evidence of common method bias.

## 4.2 Correlation analyses

Table 1 shows the results of the correlation analysis. Specifically, there is a significant positive correlation between childhood psychological abuse and cyberbullying (r = 0.398, p < 0.01); There is a significant negative correlation between childhood psychological abuse and both self-efficacy (r = -0.162, $p < 0.01$); Childhood psychological abuse and psychological resilience established a significant negative relationship (r = -0.445, $p < 0.01$); Self-efficacy was significantly and negatively related to adolescent psychological resilience (r = 0.459, $p < 0.01$); Self-efficacy was significantly and negatively related to adolescent cyberbullying(r = -0.309, $p < 0.01$); Psychological resilience was significantly and negatively related to adolescent cyberbullying(r = -0.490, $p < 0.01$). Among these correlations, the highest correlation is observed between psychological resilience and cyberbullying, while the lowest correlation is observed between childhood psychological abuse and self-efficacy.

## 4.3 Measurement model

The fit indices for the measurement model were assessed to examine how well the model fits the data. Jackson et al. have suggested that a model fits the data when the goodness-of-fit index is between 1 and 3 for $x^2$ / df, greater than 0.9 for GFI, AGFI, NFI, TLI, and CFI, less than 0.08 for SMSEA [54]. Childhood psychological abuse showed a good model fit: χ2/df = 2.939 ($X^2$ = 567.167 df = 193), RMSEA = 0.047, TLI = 0.970, NFI = 0.966, CFI = 0.977, GFI = 0.946, AGFI = 0.923. Self-efficacy showed a good model fit: χ2/df = 2.847 ($X^2$ = 54.093, df = 19), RMSEA = 0.046, TLI = 0.986, NFI = 0.991, CFI = 0.994, GFI = 0.988, AGFI = 0.964). Psychological resilience also meets the requirement with χ2/df = 3.097 ($X^2$ = 607.072, df = 196), RMSEA = 0.049, TLI = 0.962, NFI = 0.969, CFI = 0.979, GFI = 0.951, AGFI = 0.905, together with cyberbullying (χ2/df = 2.996, X2 = 245.708, df = 82, RMSEA = 0.047, TLI = 0.983, NFI = 0.989, CFI = 0.993, GFI = 0.974, AGFI = 0.933). All the data support the robustness of the measurement model.

Additionally, in the measurement model, the standardized factor loadings are significant and ideally above 0.50, indicating that the items are good indicators of their respective constructs [65]. The values of Cronbach's α and CR are over 0.7, indicating the acceptable reliability [66]. The AVE values surpassed the recommended threshold of 0.5, signifying satisfactory convergent validity, and the AVE value reaching 0.36 shows acceptable convergent validity

**Table 2. Evaluation of reliability and validity.**

| Latent variable | SC | P-value | Cronbach's a | CR | AVE | MaxR(H) |
|---|---|---|---|---|---|---|
| Childhood psychological abuse (CPA) | 0.564–0.810 | *** | 0.953 | 0.955 | 0.482 | 0.959 |
| Self-efficacy (SE) | 0.613–0.830 | *** | 0.931 | 0.932 | 0.582 | 0.939 |
| Psychological resilience (PR) | 0.528–0.818 | *** | 0.962 | 0.963 | 0.490 | 0.965 |
| Cyberbullying (CB) | 0.700–0.890 | *** | 0.974 | 0.975 | 0.660 | 0.977 |

**Table 3. The test for discriminant validity of potential variables.**

| Potential variable | Childhood psychological abuse | Self-efficacy | Psychological resilience | Cyberbullying |
|---|---|---|---|---|
| Childhood Psychological abuse | **0.694** | | | |
| Self-efficacy | -0.162*** | **0.763** | | |
| Psychological resilience | -0.447*** | 0.474*** | **0.700** | |
| Cyberbullying | 0.406*** | -0.325*** | -0.502*** | **0.812** |

Note: The square root of the AVE of four latent constructs is given in the diagonal, and the correlation coefficient is given on the below diagonal.

[67]. The square root of the AVE should be greater than the correlations with other constructs, indicating that the constructs have discriminant validity [68].

As presented in Table 2, the value of Cronbach's α ranged from 0.931 to 0.974, indicating high reliability. The standardized factor loadings covered a range between 0.528 and 0.890 ($p < .001$), while the values of CR and AVE ranged from 0.932 to 0.975 and from 0.482 to 0.660 respectively, indicating acceptable convergent validity. In Table 3, the square root of AVE for each construct was greater than the correlation with other constructs, indicating acceptable levels of discriminant validity.

## 4.4 Structural model

The structural model was evaluated using the goodness-of-fit indices and path coefficients. The fit indices for the structural model are as follows: $X^2 / df = 1.403$ ($X^2 = 1135.419$, df = 809), GFI = 0.913, AGFI = 0.901, CFI = 0.973, TII = 0.971, NFI = 0.913, RMSEA = 0.033. All the values met the recommended thresholds [54], indicating a good fit for the structural model. Additionally, as shown in Fig 2, all the path coefficients were statistically significant ($P < 0.01$) by

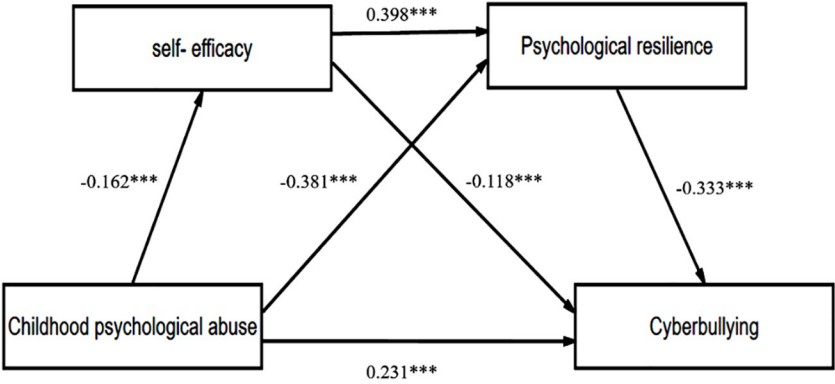

**Fig 2. The path diagram.**

performing a bootstrap procedure with 5000 resamplings. Therefore, the structural model was supported by these empirical data.

## 4.5 Testing for mediation effect

The study employed structural equation modeling to examine the mediating effects among the four variables. The bootstrap proposed by MacKinnon [69] was used for significance testing, with a sample size of 5000 and a confidence level of 95%. A mediating effect is considered statistically significant when the bootstrap 95% confidence interval of the indirect effects estimated by the bias-corrected percentile method does not include zero [69]. Data analysis was performed using Amos 24.0 software. The results of the mediation analysis for the mediating effect of self-efficacy and psychological resilience on the relationship between childhood psychological abuse and cyberbullying are presented in Table 4. The direct effect of childhood psychological abuse on adolescent cyberbullying is significant ($\beta = 0.296$, $P < 0.001$), supporting the acceptance of H1. Self-efficacy and psychological resilience mediate the relationship between childhood psychological abuse and cyberbullying, with a total indirect effect of 0.214 ($P < 0.001$). Specifically, the indirect effect is composed of three pathways: The pathway of childhood psychological abuse → self-efficacy→ cyberbullying had an indirect effect of 0.025 with a 95% confidence interval of [0.007, 0.053]; The pathway of childhood psychological abuse → self-efficacy→ psychological resilience → cyberbullying had an indirect effect of 0.028 with a 95% confidence interval of [0.013, 0.049]; The pathway of childhood psychological abuse → psychological resilience → cyberbullying had an indirect effect of 0.162 with a 95% confidence interval of [0.112, 0.227]. The Bootstrap 95% confidence intervals for all three indirect effects do not include zero, indicating that all three indirect effects are statistically significant. These results provide support for H 2, H3, and H4.

In addition, the indirect effect percentage of self-efficacy and psychological resilience as partial mediators were examined. As indicated in Table 4, among the three significant indirect mediators, the indirect effect of self-efficacy accounts for 11.5% of the total indirect effect, while the indirect effect of psychological resilience accounts for 75.7% of the total indirect

**Table 4. Mediating effects of self- efficacy and psychological resilience (N = 891).**

| Path relationship | | Effect | SE | 95% confidence internal | |
|---|---|---|---|---|---|
| | | | | Lower | upper |
| **Test of indirect, direct and total effects** | | | | | |
| DistalIE | CPA→SE→PR→CB | 0.028 | 0.009 | 0.013 | 0.049 |
| SEIE | CPA→SE→CB | 0.025 | 0.011 | 0.007 | 0.053 |
| PRIE | CPA→PR→LE | 0.162 | 0.029 | 0.112 | 0.227 |
| TIE | Total indirect effect | 0.214 | 0.034 | 0.154 | 0.289 |
| DE | CPA→CB | 0.296 | 0.065 | 0.168 | 0.423 |
| TE | Total effect | 0.510 | 0.061 | 0.392 | 0.630 |
| **Percentage of indirect effects** | | | | | |
| P1 | DistalIE/TIE | 0.128 | 0.032 | 0.068 | 0.194 |
| P2 | SEIE/TIE | 0.115 | 0.053 | 0.032 | 0.237 |
| P3 | PRIE/TIE | 0.757 | 0.069 | 0.609 | 0.882 |
| P4 | TIE/TE | 0.420 | 0.079 | 0.289 | 0.600 |
| P5 | DE/TE | 0.580 | 0.079 | 0.400 | 0.711 |

Note: CPA = Childhood psychological abuse, SE = Self-efficacy, PR = psychological Resilience, CB = cyberbullying, Standardized estimating of 5000 bootstrap sample, ***$p < 0.001$

effect. Besides, the indirect effect of self-efficacy and psychological resilience accounts for 12.8% of the total indirect effect. This indicates that the indirect effect of psychological resilience is the greatest. The specific pathways of childhood psychological abuse acting on cyberbullying through self-efficacy and psychological resilience are detailed in Fig 2.

## 5. Discussion

Empirical evidence suggests that childhood psychological abuse, self-efficacy, and psychological resilience have an impact on cyberbullying. However, there is still a gap in understanding the specific mechanisms through which childhood psychological abuse impacts cyberbullying via self-efficacy and psychological resilience. This research aimed to construct a mediation model to investigate whether childhood psychological abuse would be indirectly correlated with adolescents' cyberbullying through self-efficacy and psychological resilience. The findings, limitations and implications are presented as follows.

### 5.1 Findings

The results of the study revealed a direct and positive link between childhood psychological abuse and adolescents' cyberbullying. This not only corroborates Kircaburun et al.'s research [70], which identified a positive correlation between childhood psychological abuse and adolescents' cyberbullying but also aligns with the notion proposed by Zhang et al. [30] that psychological abuse contributes to the occurrence of cyberbullying. One potential explanation is that individuals who have experienced abuse may struggle to regulate their emotions, increasing the likelihood of displaying aggressive behavior in online settings. Adolescents who experienced greater psychological abuse during childhood are more inclined to exhibit negative online behaviors [19]. This study further underscores the significance of childhood psychological abuse as a predictive factor for cyberbullying.

The results of the study identified self- efficacy as one significant partial mediating role between childhood psychological abuse and adolescents' cyberbullying. This finding is consistent with previous research suggesting a negative association between childhood psychological abuse and self-efficacy [35, 38], as well as a negative association between self-efficacy and cyberbullying [23, 24]. These findings provide support for the idea that childhood psychological abuse plays a crucial role in shaping the perception of self-efficacy, which subsequently influences adolescents' engagement in cyberbullying behaviors. This finding adds further evidence to the understanding of the role of self-efficacy in the link between childhood psychological abuse and cyberbullying.

The results of the study demonstrated that psychological resilience plays a significant partial mediating role between childhood psychological abuse and adolescents' cyberbullying. This finding is consistent with previous research suggesting a negative association between childhood psychological abuse and psychological resilience [44, 45], as well as a negative association between psychological resilience and cyberbullying [20, 48]. One potential reason is that childhood psychological abuse can lead to a sense of helplessness, frustration, and negative emotions in children, hindering the development of psychological resilience. Individuals with lower psychological resilience may have difficulty seeking help when facing adversity and may resort to negative behaviors to avoid problems, leading to an increase in cyberbullying. These findings provide support for the idea that childhood psychological abuse plays a crucial role in shaping the perception of psychological resilience, which subsequently influence adolescents' engagement in cyberbullying behaviors.

The results of the study further showed that both self-efficacy and psychological resilience functioned as a chain mediating role between childhood psychological abuse and adolescents'

cyberbullying. In other words, adolescents with high childhood psychological abuse scores tend to perceive lower self-efficacy, leading to an overall lower belief in their ability to effectively cope with and overcome challenges. This, in turn, is associated with lower levels of psychological resilience, resulting in increased engagement in cyberbullying behaviors. This finding further elucidates the mechanisms by which environmental systems and individual factors influence adolescents' cyberbullying and advances the previous research by shedding light on how childhood psychological abuse can increase adolescents' cyberbullying. It is worth noting that although both serial mediation and self-efficacy as mediators were established, their percentages were only 12.8% and 11.5%, respectively, which were lower than the mediating effect of psychological resilience. This indicates that psychological resilience has a more significant impact on cyberbullying behaviors. This suggests that when intervening in adolescent cyberbullying behaviors at the family level, cultivating their perception of psychological resilience should be given greater priority compared to enhancing their self-efficacy.

## 5.2 Implications

The findings of this study have significant implications for both theory and practice in understanding and addressing adolescents' cyberbullying.

From a theoretical perspective, this study contributes to the existing literature by unravelling the intricate relationship between childhood psychological abuse and adolescent cyberbullying with the application of the Social Cognitive Theory. By identifying self-efficacy and psychological resilience as pivotal mediators, the study provides a conceptual framework that enhances our comprehension of the psychological processes underpinning cyberbullying behaviors. This understanding is crucial for developing psychological interventions and educational programs aimed at bolstering self-efficacy and fostering resilience among adolescents. Moreover, the findings of the study offer insights into the buffering effects of positive psychological attributes against the adverse outcomes of childhood maltreatment, enriching the existing literature on the subject and guiding future research endeavors in the field of developmental psychology and educational studies.

On a practical level, these findings offer valuable insights for designing effective interventions to prevent and address cyberbullying among adolescents. Specifically, by addressing childhood psychological abuse, enhancing self-efficacy, and fostering psychological resilience, we can reduce the likelihood of adolescents engaging in or being affected by cyberbullying. To address childhood psychological abuse, parents need to increase self-awareness and understand the impact of their emotions and behaviors on their children. They can learn positive parenting techniques such as active listening, respect, and expressing love. Additionally, establishing a positive parent-child relationship, including positive communication and emotional support, as well as clear rules and boundaries, can help reduce the occurrence of psychological abuse [71]. In enhancing self-efficacy, both schools and parents play crucial roles. Schools can design tasks that are challenging yet fair, enabling adolescents to experience success and bolster their sense of self-efficacy. Teachers should complement this by offering timely recognition and encouragement, nurturing greater confidence in their abilities. Meanwhile, parents should lead by example, exhibiting positive and proactive attitudes and behaviors. By doing so, they create an environment that allows adolescents to observe and imitate these behaviors, providing them with opportunities to practice and excel in various tasks, thereby, contributing to the development of their self-efficacy. To foster psychological resilience, parents should assist children in cultivating positive values and building self-confidence. Schools should prioritize student growth and development by establishing appropriate evaluation systems and

avoiding excessive competition. Students themselves should strive to establish positive inter-personal relationships with their peers, fostering mutual support and respect.

## 5.3 Limitations

It is important to recognize several limitations inherent in this study. Firstly, the use of a cross-sectional design precludes the establishment of causal relationships between variables. It is recommended that future research employ longitudinal or experimental designs to validate the causal hypotheses. Secondly, the reliance on self-reported data from middle school students introduces the possibility of biases, such as social desirability. Future studies should consider gathering data from multiple sources, such as parents or peers, to enhance the robustness of findings. Lastly, there are other unexplored factors in this study, such as self-control and self-esteem, which could potentially mediate the relationship. Future studies should focus on investigating the role of these factors in developing targeted interventions to reduce the occurrence of cyberbullying among adolescents.

## 6. Conclusion

The findings of this study can be summarized as follows: (1) Childhood psychological abuse, self-efficacy, psychological resilience, and cyberbullying are significantly correlated with each other. Specifically, childhood psychological abuse is significantly positively correlated with cyberbullying, while self-efficacy and psychological resilience are significantly negatively correlated with cyberbullying; (2) Childhood psychological abuse influences cyberbullying indirectly through self-efficacy and psychological resilience respectively; (3) Childhood psychological abuse can affect cyberbullying through the mediating chain role of self-efficacy and psychological resilience.

## Supporting information

**S1 Data.**
(XLSX)

## Acknowledgments

The authors wish to thank Jingtao Wu for providing technical support in data analysis for this research.

## Author Contributions

**Investigation:** Yang Han.

**Writing – original draft:** Yang Han.

**Writing – review & editing:** Haihua Ying.

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
