## [Decision Letter · Decision Letter 0]

10 Jun 2024

PONE-D-24-04494How childhood psychological abuse affects adolescent cyberbullying: The chain mediating role of self-efficacy and psychological resiliencePLOS ONE

Dear Dr. Ying,

Thank you for submitting your manuscript to PLOS ONE. After careful consideration, we feel that it has merit but does not fully meet PLOS ONE’s publication criteria as it currently stands. Therefore, we invite you to submit a revised version of the manuscript that addresses the points raised during the review process.

We look forward to receiving your revised manuscript.

Kind regards,

Amgad Muneer

Academic Editor

PLOS ONE

Journal Requirements:

[46301]. 

3. In the online submission form, you indicated that [Data are available from the Hohai University Ethics Committee (contact via Haihua Ying) for researchers who meet the criteria for access to confidential data.]. 

Reviewers' comments:

Reviewer's Responses to Questions

**Comments to the Author**

1. Is the manuscript technically sound, and do the data support the conclusions?

Reviewer #1: Yes

Reviewer #2: Yes

2. Has the statistical analysis been performed appropriately and rigorously? 

Reviewer #1: No

Reviewer #2: N/A

3. Have the authors made all data underlying the findings in their manuscript fully available?

Reviewer #1: Yes

Reviewer #2: Yes

4. Is the manuscript presented in an intelligible fashion and written in standard English?

Reviewer #1: Yes

Reviewer #2: Yes

5. Review Comments to the Author

Reviewer #1: Review of Manuscript PONE-D-24-04494

Dear Authors,

Firstly, I would like to congratulate you on your significant contribution to the literature through this insightful article. The topic of understanding the underlying mechanisms linking childhood psychological abuse to adolescent cyberbullying is indeed crucial. Your work sheds light on potential pathways for targeted interventions and support programs aimed at promoting the well-being of adolescents facing early adversity.

Introduction:

While the introduction provides a clear argument about the study context and literature gaps, it can be further enhanced by addressing the following key points:

Positioning the Study:

The introduction should position the study within the broader context of existing literature. This involves not only identifying the gaps but also explaining why it is necessary to explore the specific objectives of this study. Highlight the importance of the study by discussing the practical and theoretical implications briefly in the introduction section. Explain how this research can contribute to the existing body of knowledge and why it is critical to understand the mechanisms linking childhood psychological abuse to cyberbullying.

Literature Gaps:

Provide a more in-depth discussion on the specific gaps in the literature that this article aims to address. Detail how your research will close some of these gaps, contributing to the field in a novel way.

Research Questions (Objectives) and Contributions and Beneficiaries:

Clearly state the research questions (RQs) and outline the principal contributions of the study. This helps in setting the expectations for the readers and provides a roadmap for the rest of the article.

Discuss the objectives of the study explicitly and identify the primary beneficiaries. For instance, mention how authorities, parents, educators, and policymakers can benefit from the findings of this research.

Article Structure:

Include a brief overview of the article’s structure at the end of the introduction. This should give the readers an idea of what to expect in the subsequent sections, making it easier for them to follow the narrative.

In summary, the introduction should provide a comprehensive snapshot of the entire article. It should encapsulate the study’s significance, objectives, research questions, literature gaps, and potential beneficiaries. An effective introduction captures the reader's attention and compels them to read further, providing them with a clear understanding of what the article entails.

Theoretical Background and Hypothesis Development:

To enhance the clarity and depth of the theoretical framework, I suggest renaming this section to "Theoretical Background and Hypothesis Development." This section should comprehensively cover the approaches and theories relevant to the article's context, along with a precise justification for the selection of the studied variables and their explicit definitions. Below are specific recommendations for improving this section:

Discussion of the Theory and its importance:

Begin with a thorough discussion of the chosen theory, highlighting its significance in enhancing our understanding of the topic. Explain why this particular theory is more suitable compared to other theories in the context of this study. This will provide readers with a strong rationale for the theoretical foundation of your research.

Elaborate on the importance of the Social Cognitive Theory (SCT) in understanding cyberbullying behaviors. Discuss its relevance and how it helps in explaining the mechanisms linking childhood psychological abuse to cyberbullying through self-efficacy and psychological resilience.

In the following points, researchers have written some of the parts I have mentioned in my comments, so they should read my comments and add the missing parts only:

Justification of Variable Selection:

Provide a more precise justification for selecting the studied variables, such as childhood psychological abuse, self-efficacy, psychological resilience, and cyberbullying. Explicitly define each variable and explain its relevance to the study. This helps in establishing a clear connection between the theory and the research variables.

Definition and Impact of Childhood Psychological Abuse:

Enhance the discussion on childhood psychological abuse by incorporating insights from psychology literature. Explain how childhood psychological abuse influences psychological development and brain function. This can include references to studies on the neurological and psychological impacts of such abuse, providing a more rigorous and enriched discussion.

Linking Theory to Hypothesis Development:

After establishing the theoretical foundation and defining the variables, proceed with discussing the relationships that lead to hypothesis development. Ensure that each hypothesis is clearly derived from the theoretical framework and supported by relevant literature. This logical flow will strengthen the argumentation and coherence of your hypothesis development.

By restructuring and enriching this section, you will provide a more robust theoretical foundation for your study. It will also make the rationale for your hypotheses clearer and more compelling, thus enhancing the overall quality of your manuscript.

Methodology:

The methodology section is crucial for validating the robustness and reliability of the study. Below are my suggestions for enhancing this section:

Justification of Participant Number:

Provide a detailed justification for the total number of participants. Explain how the number of 891 participants was determined. This should include considerations such as the study’s objectives, the expected effect size, and the desired power of the study.

Sampling Design:

Elaborate on the sampling design used in the study. Describe the sampling method (e.g., random sampling, stratified sampling) and justify why this method was chosen. Include information about how participants were selected from the four public secondary schools in Jiangsu Province and how this sampling design helps achieve a representative sample of the population.

Sample Size Determination:

Discuss the methods used to determine the sample size before conducting the study. This could include power analysis or other statistical methods to ensure that the sample size is adequate for detecting the expected effects. Provide details on the calculations or criteria used to arrive at the required sample size.

Data Analysis Process:

Clearly describe the data analysis process and procedures used in the study without presenting the results in this section. Explain the steps taken for data cleaning, handling missing data, and the statistical techniques used for analyzing the data (e.g., descriptive statistics, reliability testing, Pearson correlation analysis, structural equation modeling). Discuss the rationale for choosing these techniques and how they align with the study’s objectives.

Relocation of Results:

Move the results currently presented in the "2.3 Questionnaire Design" section to the Results section. This ensures a clear distinction between the methodology and the findings of the study. The methodology should focus on how the data was collected and analyzed, while the Results section should present the outcomes of these analyses.

By incorporating these elements, the methodology section will provide a clearer, more comprehensive explanation of the study’s design and execution. This will help in establishing the credibility and reliability of the research findings.

Results:

Before delving into the assessment of the structural equation model, it is essential to discuss the assessment of the measurement model. This involves evaluating the model fit, factor loadings, reliability, and validity of the constructs used in the study.

Assessment of Measurement Model:

Model Fit:

Report the fit indices for the measurement model, including chi-square (χ²), degrees of freedom (df), chi-square to degrees of freedom ratio (χ²/df), Root Mean Square Error of Approximation (RMSEA), Tucker-Lewis Index (TLI), Comparative Fit Index (CFI), Normed Fit Index (NFI), Goodness of Fit Index (GFI), and Adjusted Goodness of Fit Index (AGFI). These indices provide an overall assessment of how well the model fits the data.

Factor Loadings:

Present the factor loadings for each item. Factor loadings should be significant and ideally above 0.50, indicating that the items are good indicators of their respective constructs.

Reliability:

Cronbach's α: Report the Cronbach's alpha values for each construct to assess internal consistency. Values above 0.70 are generally considered acceptable.

Composite Reliability (CR): Provide the composite reliability values for each construct. CR values above 0.70 indicate good reliability.

MaxR(H): Discuss the MaxR(H) values, which provide an additional measure of construct reliability.

Validity:

Average Variance Extracted (AVE): Report the AVE values for each construct. AVE values above 0.50 suggest that the construct explains more than half of the variance of its indicators.

Discriminant Validity: Ensure discriminant validity by comparing the square root of the AVE for each construct with the inter-construct correlations. The square root of the AVE should be greater than the correlations with other constructs, indicating that the constructs are distinct from each other.

By providing a thorough assessment of the measurement model, the study can demonstrate the reliability and validity of the constructs used in the analysis. This foundational step ensures that the subsequent assessment of the structural equation model is based on a solid measurement framework.

Assessment of Structural Equation Modeling:

Proceed with the assessment of the structural equation model. This includes testing the hypothesized relationships between childhood psychological abuse, self-efficacy, psychological resilience, and cyberbullying.

Present the path coefficients, significance levels, and fit indices for the structural model.

Discuss the indirect, and total effects as hypothesized in the study. Use bootstrap analysis to confirm the significance of indirect effects.

By structuring the results section in this manner, the study provides a clear and comprehensive evaluation of both the measurement and structural models, ensuring the robustness and reliability of the findings.

Findings and Discussion:

The findings and discussion sections of the manuscript are sound. However, there should be a more detailed discussion on the theoretical contributions and how this research enhances existing knowledge. Additionally, the limitations section should be moved to the end, just before the conclusion. Furthermore, the article would benefit from a concluding remarks section that effectively summarizes the key points of the study.

Reviewer #2: Thank you for allowing me to review this research.

This research provides a compelling analysis of how childhood psychological abuse affects adolescent cyberbullying: The chain mediating role of self-efficacy and psychological resilience.

This work significantly contributes to the literature, underscoring the critical roles of usability, utility, and trust in technology acceptance. However, I have to admit that several concerns arose while reading the paper. Below, I elaborate on these issues in the hope that they will help the authors further strengthen the manuscript:

1. The authors should address the literature gap more.

2. The theoretical foundation for the four research hypotheses appears overly concise. Specifically, the explanation of the moderating effect in the fourth hypothesis needs further elaboration and strengthening.

3. The authors should provide more specific operational definitions and the theoretical foundations for the five research dimensions.

4. The authors should include the complete questionnaire items for the five research dimensions and explain their theoretical foundations.

5. The authors should present the test results for common method bias.

Overall, I hope these comments will help the authors in their future efforts to revise the manuscript. Furthermore, I would like to encourage the authors to continue their efforts in this area.

6. PLOS authors have the option to publish the peer review history of their article (what does this mean?). If published, this will include your full peer review and any attached files.

Reviewer #1: No

Reviewer #2: No

---

## [Author Response · Author response to Decision Letter 0]

6 Aug 2024

We would like to express our gratitude to the editor and reviewers for their valuable input, which has undoubtedly improved the quality of our manuscript. We studied the comments carefully and made corrections as suggested. Point-by-point responses to the comments of each reviewer are as following: 

Response to Reviewer 1

Introduction

1.Reviewer’s comment: Positioning the Study: The introduction should position the study within the broader context of existing literature. This involves not only identifying the gaps but also explaining why it is necessary to explore the specific objectives of this study. Highlight the importance of the study by discussing the practical and theoretical implications briefly in the introduction section. Explain how this research can contribute to the existing body of knowledge and why it is critical to understand the mechanisms linking childhood psychological abuse to cyberbullying.

Our answer: Thank you for your valuable feedback and suggestion. We have added the context of existing literature. The added content in red can be found in the first paragraph of the introduction. Besides, we have identified the gaps and explained why it is necessary to explore the specific objectives of this study，which can found in the second paragraph on Page 3. Lastly, we have highlighted the importance of the study “This study is significant as it addresses a gap in the existing literature and provides insights and guidance for educators and policymakers seeking to reduce adolescent cyberbullying” in the second paragraph on Page 3.

2.Reviewer’s comment: Literature Gaps: Provide a more in-depth discussion on the specific gaps in the literature that this article aims to address. Detail how your research will close some of these gaps, contributing to the field in a novel way.

Our answer: Thank you for your valuable feedback and suggestion. The more discussion has been added on the specific gaps in the literature on Page 3 “this study aims to investigate the interactive effects of childhood psychological abuse, self-efficacy, psychological resilience on adolescent cyberbullying, thereby providing a holistic understanding of the relationship between these factors. Furthermore, the study endeavors to investigate the impact of childhood psychological abuse on adolescent cyberbullying, with a specific focus on the mediating roles of self-efficacy and psychological resilience.” Besides, details about significance have been added: “This study is significant as it addresses a gap in the existing literature and provides insights into the determinants of adolescent cyberbullying. Moreover, by exploring the mediating mechanisms through which childhood psychological abuse impacts adolescent cyberbullying, this study provides valuable guidance for educators and parents seeking to reduce adolescent cyberbullying.” on Page 3.

3.Reviewer’s comment: Research Questions (Objectives) and Contributions and Beneficiaries: Clearly state the research questions (RQs) and outline the principal contributions of the study.

This helps in setting the expectations for the readers and provides a roadmap for the rest of the article.

Our answer: Thank you for your valuable feedback and suggestion. The four research questions have been added on Page 3: “This study seeks to address the following questions: First, what is the relationship between childhood psychological abuse and adolescent cyberbullying? Second, does self-efficacy mediate the relationship between childhood psychological abuse and adolescent cyberbullying? Third, does psychological resilience mediate the relationship between childhood psychological abuse and adolescent cyberbullying? Fourth, is there a serial mediation effect of self-efficacy and psychological resilience between childhood psychological abuse and adolescent cyberbullying?”

4.Reviewer’s comment: Discuss the objectives of the study explicitly and identify the primary beneficiaries. For instance, mention how authorities, parents, educators, and policymakers can benefit from the findings of this research. 

Our answer: Thank you for your valuable feedback and suggestion. We have discussed the objectives of the study explicitly and identify the primary beneficiaries。The objective can be found on Page 3.

5.Reviewer’s comment: Article Structure: Include a brief overview of the article’s structure at the end of the introduction. This should give the readers an idea of what to expect in the subsequent sections, making it easier for them to follow the narrative. 

Our answer: Thank you for your valuable feedback and suggestion. The structure of the article has been presented on Page 3: The structure of the remaining sections of this article is as follows. Section 2 provides an overview of the theoretical background and hypothesis development. Section 3 details the materials and methods, encompassing participants, the research process, research tools, and statistical analysis. Section 4 covers common method variance, descriptive statistics, correlation analysis, examination of the model, and testing for mediation effects. Section 6 presents the findings, limitations, and implications.

6.Reviewer’s comment: I suggest renaming this section to "Theoretical Background and Hypothesis Development."

Our answer: Thank you for your valuable feedback and suggestion. We have renamed the section of “2. Theoretical Background and Hypothesis Development” on Page 3. 

7.Reviewer’s comment: Discussion of the Theory and its importance: Begin with a thorough discussion of the chosen theory, highlighting its significance in enhancing our understanding of the topic. Explain why this particular theory is more suitable compared to other theories in the context of this study. This will provide readers with a strong rationale for the theoretical foundation of your research. Besides, elaborate on the importance of the Social Cognitive Theory (SCT) in understanding cyberbullying behaviors. Discuss its relevance and how it helps in explaining the mechanisms linking childhood psychological abuse to cyberbullying through self-efficacy and psychological resilience. 

 Our answer: Thank you for your valuable feedback and suggestion. We have highlighted the significance of SCT, and explain why this particular theory is more suitable Please see the content in the section of ”2.1 Theoretical background”. We have also elaborated on the importance of the Social Cognitive Theory (SCT) in understanding cyberbullying behaviors, and discuss its relevance and how it helps in explaining the mechanisms. Please read the red letter in the section of’ “2.1 Theoretical background” on Page 3.

8.Reviewer’s comment: Justification of Variable Selection: Provide a more precise justification for selecting the studied variables, such as childhood psychological abuse, self-efficacy, psychological resilience, and cyberbullying. Explicitly define each variable and explain its relevance to the study. This helps in establishing a clear connection between the theory and the research variables. 

Our answer: Thank you for your valuable feedback and suggestion. The four variables have 

been defined. For example, “Cyberbullying is characterized by deliberate, repetitive, and malicious acts which are carried out using modern communication technologies, aimed at causing harm to others [2-3]. It comprises two dimensions: cyberbullying victimization and cyberbullying perpetration [50].” on Page 1. “Child psychological abuse refers to the series of inappropriate fostering methods that are repeatedly and continuously adopted by the fosterer during the process of children’s growth, including intimidation, neglect, disparagement, interference, and indulgence” on Page 2； “Self-efficacy refers to an individual’s confidence and expectation in their ability to take effective action and accomplish tasks in specific situations” on the top of Page 5, and “Psychological resilience is defined as the adaptive ability to maintain an active life despite adversity and stressful events” on the bottom of Page 5.

9.Reviewer’s comment: Definition and Impact of Childhood Psychological Abuse: Enhance the discussion on childhood psychological abuse by incorporating insights from psychology literature. Explain how childhood psychological abuse influences psychological development and brain function. This can include references to studies on the neurological and psychological impacts of such abuse, providing a more rigorous and enriched discussion.

Our answer: Thank you for your valuable feedback and suggestion. We have added the literature about how childhood psychological abuse influences psychological development and brain function. Please read the section of “2.2.1 Childhood psychological abuse and cyberbullying” on Page 4.

 10. Reviewer’s comment: Linking Theory to Hypothesis Development: After establishing the theoretical foundation and defining the variables, proceed with discussing the relationships that lead to hypothesis development. Ensure that each hypothesis is clearly derived from the theoretical framework and supported by relevant literature. This logical flow will strengthen the argumentation and coherence of your hypothesis development. By restructuring and enriching this section, you will provide a more robust theoretical foundation for your study. It will also make the rationale for your hypotheses clearer and more compelling, thus enhancing the overall quality of your manuscript. 

Our answer: Thank you for your valuable feedback and suggestion. We have made revisions, which can be found in the section of “2. Theoretical Background and Hypothesis Development.” on Page 3

Methodology

11.Reviewer’s comment: Justification of Participant Number: Provide a detailed justification for the total number of participants. Explain how the number of 891 participants was determined. This should include considerations such as the study’s objectives, the expected effect size, and the desired power of the study.

Our answer: Thank you for your valuable feedback and suggestion. We have provided a method of determining the total number of participants. Please read the section of “3.1 Participants” on Page 7：“This study utilized G*power 3.1 software (Faul et al., 2007) to calculate the required sample size, with an effect size set at 0.3 and α set at 0.05. The results indicated that in order to achieve a statistical power of 0.95, a total of 145 participants were needed. Furthermore, based on the requirement of Structural Equation Modeling (SEM) (Zhang et al., 2020) that the appropriate sample size should be at least ten times the total observed variables, it was determined that a minimum of 800 participants would be necessary.”

12.Reviewer’s comment: Elaborate on the sampling design used in the study. Describe the sampling method (e.g., random sampling, stratified sampling) and justify why this method was chosen. Include information about how participants were selected from the four public secondary schools in Jiangsu Province and how this sampling design helps achieve a representative sample of the population. 

Our answer: Thank you for your valuable feedback and suggestion. We have described the sampling method, and justify why this method was chosen, and Include information about how participants were selected. Please read the red content “The survey initially identified schools for sample collection based on convenience sampling principles. However, to ensure representativeness, cluster sampling was subsequently employed at the class level to select the 1,000 samples from 4 secondary schools (2 public junior high schools and 2 public senior high schools) in Jiangsu province, China.” in the part of “3.1 Participants” on Page 7.

13.Reviewer’s comment: Sample Size Determination: Discuss the methods used to determine the sample size before conducting the study. This could include power analysis or other statistical methods to ensure that the sample size is adequate for detecting the expected effects. Provide details on the calculations or criteria used to arrive at the required sample size.

Our answer: Thank you for your valuable feedback and suggestion. We have discussed the methods used to determine the sample size，include power analysis，and provide details on the calculations or criteria used to arrive at the required sample size. They are presented in the part of “3.1 Participants” on Page 7. 

14. Reviewer’s comment: Data Analysis Process: Clearly describe the data analysis process and procedures used in the study without presenting the results in this section. Explain the steps taken for data cleaning, handling missing data, and the statistical techniques used for analyzing the data (e.g., descriptive statistics, reliability testing, Pearson correlation analysis, structural equation modeling). Discuss the rationale for choosing these techniques and how they align with the study’s objectives.

Our answer: Thank you for your valuable feedback and suggestion. We have clearly described the data analysis process and procedures on Page 7. 

15. Reviewer’s comment: Relocation of Results: Move the results currently presented in the “2.3 Questionnaire Design” section to the Results section. This ensures a clear distinction between the methodology and the findings of the study. The methodology should focus on how the data was collected and analyzed, while the Results section should present the outcomes of these analyses.

By incorporating these elements, the methodology section will provide a clearer, more comprehensive explanation of the study’s design and execution. This will help in establishing the credibility and reliability of the research findings.

Our answer: Thank you for your valuable feedback and suggestion. We have revised the section title from “3.3 Questionnaire Design” to “3.3 Research Instrument” and have relocated the content regarding the reliability and validity of each scale used in this study to the section “4.3 Measurement Model” on Page 9.

Results:

16.Reviewer’s comment: Model Fit: Report the fit indices for the measurement model, including chi-square (χ²), degrees of freedom (df), chi-square to degrees of freedom ratio (χ²/df), Root Mean Square Error of Approximation (RMSEA), Tucker-Lewis Index (TLI), Comparative Fit Index (CFI), Normed Fit Index (NFI), Goodness of Fit Index (GFI), and Adjusted Goodness of Fit Index (AGFI). These indices provide an overall assessment of how well the model fits the data.

Our answer: Thank you for your valuable feedback and suggestion. We have reported the fit indices for the measurement model, which can be found in the section of “4.3 Measurement model” on Page 9.

 17. Reviewer’s comment: Factor Loadings: Present the factor loadings for each item. Factor loadings should be significant and ideally above 0.50, indicating that the items are good indicators of their respective constructs.

Our answer: Thank you for your valuable feedback and suggestion. We have presented the range the factor loadings for each item in the article. The factor loadings are presented in Table 2 on Page 10.

18. Reviewer’s comment: Reliability: Cronbach's α: Report the Cronbach's alpha values for each construct to assess internal consistency. Values above 0.70 are generally considered acceptable. Composite Reliability (CR): Provide the composite reliability values for each construct. CR values above 0.70 indicate good reliability. MaxR(H): Discuss the MaxR(H) values, which provide an additional measure of construct reliability.

Our answer: Thank you for your valuable feedback and suggestion. Cronbach’s α, CR, and MaxR(H) values are presented in Table 2 on Page 10. 

19.Reviewer’s comment: Validity: Report Average Variance Extracted (AVE) and discriminant Validity.

Our answer: Thank you for your valuable feedback and suggestion. The values about AVE and 

discriminant Validity are presented in Table2 and Table 3 on Page 10. 

20. Reviewer’s comment: Assessment of Structural Equation Modeling: Proceed with the assessment of the structural equation model. This includes testing the hypothesized relationships between childhood psychological abuse, self-efficacy, psychological resilience, and cyberbullying.

Present the path coefficients, significance levels, and fit indices for the structural model. Discus

---

## [Decision Letter · Decision Letter 1]

22 Aug 2024

How childhood psychological abuse affects adolescent cyberbullying: The chain mediating role of self-efficacy and psychological resilience

PONE-D-24-04494R1

Dear Dr. Ying,

We’re pleased to inform you that your manuscript has been judged scientifically suitable for publication and will be formally accepted for publication once it meets all outstanding technical requirements.

Kind regards,

Amgad Muneer

Academic Editor

PLOS ONE

**Comments to the Author**

1. If the authors have adequately addressed your comments raised in a previous round of review and you feel that this manuscript is now acceptable for publication, you may indicate that here to bypass the “Comments to the Author” section, enter your conflict of interest statement in the “Confidential to Editor” section, and submit your "Accept" recommendation.

Reviewer #1: All comments have been addressed

Reviewer #2: All comments have been addressed

2. Is the manuscript technically sound, and do the data support the conclusions?

Reviewer #1: Yes

Reviewer #2: Yes

3. Has the statistical analysis been performed appropriately and rigorously? 

Reviewer #1: Yes

Reviewer #2: Yes

4. Have the authors made all data underlying the findings in their manuscript fully available?

Reviewer #1: Yes

Reviewer #2: Yes

5. Is the manuscript presented in an intelligible fashion and written in standard English?

Reviewer #1: Yes

Reviewer #2: Yes

6. Review Comments to the Author

Reviewer #1: Thank you for addressing my previous comments and suggestions. I have reviewed the revised manuscript, and I am pleased with the improvements made. The changes have been well-executed, and I believe the manuscript is now in good shape. No further revisions are required from my side.

Congratulations on the successful revision.

Reviewer #2: (No Response)

7. PLOS authors have the option to publish the peer review history of their article (what does this mean?). If published, this will include your full peer review and any attached files.

Reviewer #1: No

Reviewer #2: No

---

## [Editor Report · Acceptance letter]

29 Aug 2024

PONE-D-24-04494R1 

PLOS ONE

Dear Dr. Ying, 

I'm pleased to inform you that your manuscript has been deemed suitable for publication in PLOS ONE. Congratulations! Your manuscript is now being handed over to our production team.

Kind regards, 

on behalf of

Dr. Amgad Muneer 

Academic Editor

PLOS ONE